# Analyzing human knockouts to validate *GPR151* as a therapeutic target for reduction of body mass index

Allan Gurtan[1], John Dominy[1], Shareef Khalid[2,3,4], Linh Vong[1], Shari Caplan[1], Treeve Currie[1], Sean Richards[1], Lindsey Lamarche[1], Daniel Denning[1], Diana Shpektor[1], Anastasia Gurinovich[1,5], Asif Rasheed[2,6], Shahid Hameed[7], Subhan Saeed[2], Imran Saleem[7], Anjum Jalal[8], Shahid Abbas[8], Raffat Sultana[9], Syed Zahed Rasheed[9], Fazal-ur-Rehman Memon[10], Nabi Shah[11], Mohammad Ishaq[9], Amit V. Khera[12], John Danesh[13], Philippe Frossard[2], Danish Saleheen[2,3,4]*

1 Novartis Institutes for BioMedical Research, Cambridge, Massachusetts, United States of America, 2 Center for Non-Communicable Diseases, Karachi, Sindh, Pakistan, 3 Department of Medicine, Columbia University Irving Medical Center, New York, New York, United States of America, 4 Department of Cardiology, Columbia University Irving Medical Center, New York, New York, United States of America, 5 Tufts Medical Center, Boston, Massachusetts, United States of America, 6 TopMed Hospital, Karachi, Sindh, Pakistan, 7 Punjab Institute of Cardiology, Lahore, Pakistan, 8 Faisalabad Institute of Cardiology, Faisalabad, Pakistan, 9 Karachi Institute of Heart Diseases, Karachi, Pakistan, 10 Red Crescent Institute of Cardiology, Hyderabad, Pakistan, 11 Department of Pharmacy, COMSATS University Islamabad, Islamabad, Pakistan, 12 Center for Genomic Medicine, Department of Medicine, Massachusetts General Hospital, Boston, Massachusetts, United States of America, 13 BHF Cardiovascular Epidemiology Unit, Department of Public Health and Primary Care, Cambridge University & Health Data Research UK, Wellcome Sanger Institute, Cambridge, United Kingdom

☯ These authors contributed equally to this work.
* ds3792@cumc.columbia.edu

**Data Availability Statement:** The whole-exome sequencing data that we have generated includes rare loss-of-function variants including many that have a count of less than 5. This could potentially

## Abstract

Novel drug targets for sustained reduction in body mass index (BMI) are needed to curb the epidemic of obesity, which affects 650 million individuals worldwide and is a causal driver of cardiovascular and metabolic disease and mortality. Previous studies reported that the Arg95Ter nonsense variant of GPR151, an orphan G protein-coupled receptor, is associated with reduced BMI and reduced risk of Type 2 Diabetes (T2D). Here, we further investigate *GPR151* with the Pakistan Genome Resource (PGR), which is one of the largest exome biobanks of human homozygous loss-of-function carriers (knockouts) in the world. Among PGR participants, we identify eleven GPR151 putative loss-of-function (plof) variants, three of which are present at homozygosity (Arg95Ter, Tyr99Ter, and Phe175LeufsTer7), with a cumulative allele frequency of 2.2%. We confirm these alleles *in vitro* as loss-of-function. We test if *GPR151* plof is associated with BMI, T2D, or other metabolic traits and find that *GPR151* deficiency in complete human knockouts is not associated with clinically significant differences in these traits. Relative to Gpr151+/+ mice, Gpr151-/- animals exhibit no difference in body weight on normal chow and higher body weight on a high-fat diet. Together, our findings indicate that GPR151 antagonism is not a compelling therapeutic approach to treatment of obesity.

lead to identification of study participants. Hence, all academic requests to access relevant data should be sent to ks76@cncdpk.com. CNCD will ask relevant investigators to sign a data confidentiality agreement which would limit any investigator not to de-identify any of the study participants.

**Funding:** D.Sa. has received grants from the National Institutes of Health (www.nih.gov) (R01-HL-145437), (R01-HG-010689), (R01-HL133339), (X01HL139399), (RC2 HL101834-01) (RC1 TW008485-01). Employees of NIBR were involved in study design, data collection and analysis, decision to publish, or preparation of the manuscript.

**Competing interests:** I have read the journal's policy and the authors of this manuscript have the following competing interests: A.Gurt., J.D., S.C., T.C., S.R., L.L., and D.D. are employees of the Novartis Institutes for BioMedical Research (NIBR). L.V. is currently an employee of Alnylam. D.Sh. is currently an employee of Bristol Myers Squibb. D.Sa. has received funding from Novartis, Regeneron Pharmaceuticals, GSK, Genentech, AstraZeneca, Novo Nordisk, NGM, Eli Lilly and Variant Bio. A.V.K. has served as a scientific advisor to Sanofi, Amgen, Maze Therapeutics, Navitor Pharmaceuticals, Sarepta Therapeutics, Verve Therapeutics, Veritas International, Color Health, Third Rock Ventures, and Columbia University (NIH); received speaking fees from Illumina, MedGenome, Amgen, and NIBR; and received a sponsored research agreement from NIBR.

## Author summary

Human genetics studies can provide compelling targets for therapeutic intervention. While some therapeutic targets, such as *PCSK9*, are based on extensive genetic validation, many others are based on weaker associations with variants of unknown consequence that require further validation. Recent publications reported associations between loss of *GPR151* function and low body mass index (BMI), raising the possibility of inhibiting *GPR151* for the treatment of obesity and metabolic syndromes. To evaluate the relationship between *GPR151* and BMI, we (1) identified and experimentally confirmed loss-of-function variants present in the Pakistan Genome Resource (PGR) biobank, one of the world's largest biobanks of human gene "knockouts", (2) analyzed these loss-of-function variants individually and in burden tests for association with BMI and other metabolic traits or diseases, and (3) verified the evolutionary conservation of our findings in mice lacking *Gpr151*. We observe that *GPR151* loss does not affect BMI to a clinically relevant extent and conclude that inhibiting *GPR151* may not be effective at treating obesity.

## Introduction

Obesity, defined as a body mass index (BMI) of $>30$ kg/m$^2$, is a major global health concern. In 2015, 7.1% of global deaths were attributable to high BMI [1]. Predictions estimate that half of the world's population will be obese by 2030 [2]. By 2030 in the United States alone, 25% of the population may be severely obese, as defined by BMI$>35$ kg/m$^2$ [3]. Being overweight or obese leads to a steep increase in all-cause mortality [4], largely through increased risk for Type 2 Diabetes (T2D) [5], nonalcoholic fatty liver disease (NAFLD) [6], and cardiovascular disease (CVD) [1]. Reduction in BMI appears to remit many obesity-associated disorders [7].

Given the mortality, morbidity and public cost associated with obesity, there is a strong interest in identifying drug targets for sustained reduction in BMI. To date, two particular therapies that promote body weight reduction also reduce hospitalization and mortality from associated co-morbidities. Glucagon-like peptide-1 (GLP-1) agonists are incretin mimetics that can reduce body weight by up to 12%, improve glycemic status, and reduce cardiovascular events [8]. SGLT2 inhibitors prevent glucose re-uptake in kidneys, reduce body weight by 2 kg, improve glycemic status, reduce cardiovascular events, and improve kidney function [9–14]. Both GLP-1 agonists and SGLT2 inhibitors modify glucose metabolism, nonetheless, their effect on body weight is consistent with the expectation that reduction of BMI is therapeutically beneficial.

Human studies have identified numerous genetic loci associated with BMI [15]. However, many of the genes linked to these loci are either technically challenging to drug or are poorly validated as causal. For example, the fat mass and obesity-associated (*FTO*) gene encodes an mRNA demethylase [16–17] strongly associated with BMI [18]. However, a direct role for *FTO* in regulating BMI is unclear and has been called into question by studies suggesting linkage to variants in nearby genes *IRX3* and *IRX5* [19–21]. Loss-of-function variants in melanocortin 4 receptor (MC4R) are associated with obesity in humans [22–24]. MC4R modulators were often associated with adverse cardiovascular side effects until the identification of setmelanotide, which does not elicit these undesirable effects and was approved in the United States and Europe for treatment of genetic obesity [25]. Identification of additional BMI-associated genes may provide greater insight into the biology of body weight control and yield genes for which therapeutic modulation is tractable.

G protein coupled receptors (GPCRs) are tractable drug targets that have been associated with numerous phenotypes in human genetics studies and in mouse models [26]. GPR151 is a poorly understood, brain-specific GPCR [27] for which loss-of-function has been associated with decreased BMI [28–30]. Additionally, at *GPR151*, carriage of rare (alternative allele frequency [aaf] < 1%) putative loss-of-function (plof) variants and bioinformatically predicted damaging missense variants have also been associated with a decrease in BMI [30].

To date, homozygous plof carriers (human knockouts) of *GPR151* have not been reported in detail. To follow up on the published genetic association, we use the Pakistan Genome Resource (PGR), which is the world's largest biobank of human homozygous plof carriers (knockouts) identified through whole-exome sequencing of >80,000 participants. Here, we (i) identify homozygous carriers of *GPR151* plof variants, including those specific to South Asia, (ii) confirm *in vitro* that these variants are loss-of-function, (iii) test if *GPR151* knockouts are associated with BMI, T2D, or other metabolic traits, and (iv) characterize *Gpr151*$^{-/-}$ mice for body weight.

## Results and discussion

### Association of *GPR151* plof variants with BMI and cardiometabolic events

The PGR at the Center for Non-Communicable Diseases (CNCD) in Pakistan is a large biobank of highly consanguineous participants. A medical history and numerous clinical measurements including BMI, T2D status, and myocardial infarction (MI) status, are available for most participants. In the PGR, we identified a total of 11 plof variants, with a cumulative allele frequency of 2.2%, including 48 homozygous carriers of three plof variants in *GPR151* (S1 Table). Variants with homozygous carriers included Arg95Ter, Tyr99Ter and Phe175LeufsTer7. The latter two variants are highly enriched in South Asia compared to other populations. In UK Biobank, 21 homozygous plof carriers were identified among 281,852 exome-sequenced participants, and in gnomAD 13 homozygous plof carriers were found in non-south-Asian populations [31].

*GPR151* is expressed from a single exon, and nonsense substitutions are more likely to escape nonsense mediated decay (NMD) in single-exon genes. To determine if truncated proteins are expressed from *GPR151* variant transgenes, we transiently transfected HEK293 cells with cDNA expression constructs corresponding to variants for which we identified homozygous plof carriers in PGR. Changes in GPR151 protein sequence may alter epitopes detected by antibodies specific to GPR151 and thus confound detection by western blot. Therefore, constructs were tagged at the N-terminus with an HA epitope tag to directly compare expression of GPR151 variants *in vitro*. From transfected cells, total cell lysates and isolated membrane extracts were generated and evaluated by western blot (Fig 1). The reference allele (i.e. wild-type [WT]) GPR151 protein expressed at high levels and was detected in both the total lysate and in membrane extracts. In contrast, Arg95Ter and Tyr99Ter were not detectable in either extract. Although, the Phe175LeufsTer7 variant protein was detectable in both the total cell and membrane lysates, migrating at a size consistent with the expected truncation, it expressed at significantly lower levels compared to wild-type, indicating impaired stability. Phe175LeufsTer7 is missing the last three of the protein's seven transmembrane domains, the entire cytoplasmic tail, and intracellular loop 3, which is typically critical for G protein activity. The severity and diminished expression of this truncation suggest that this variant is a loss-of-function. Our *in vitro* observations confirm that the homozygous *GPR151* plof variants in PGR are loss-of-function alleles.

To test for associations with BMI, we analyzed plof variants individually and in a gene burden test to increase power. For our gene-burden analyses (cumulative allele frequency [caf] =

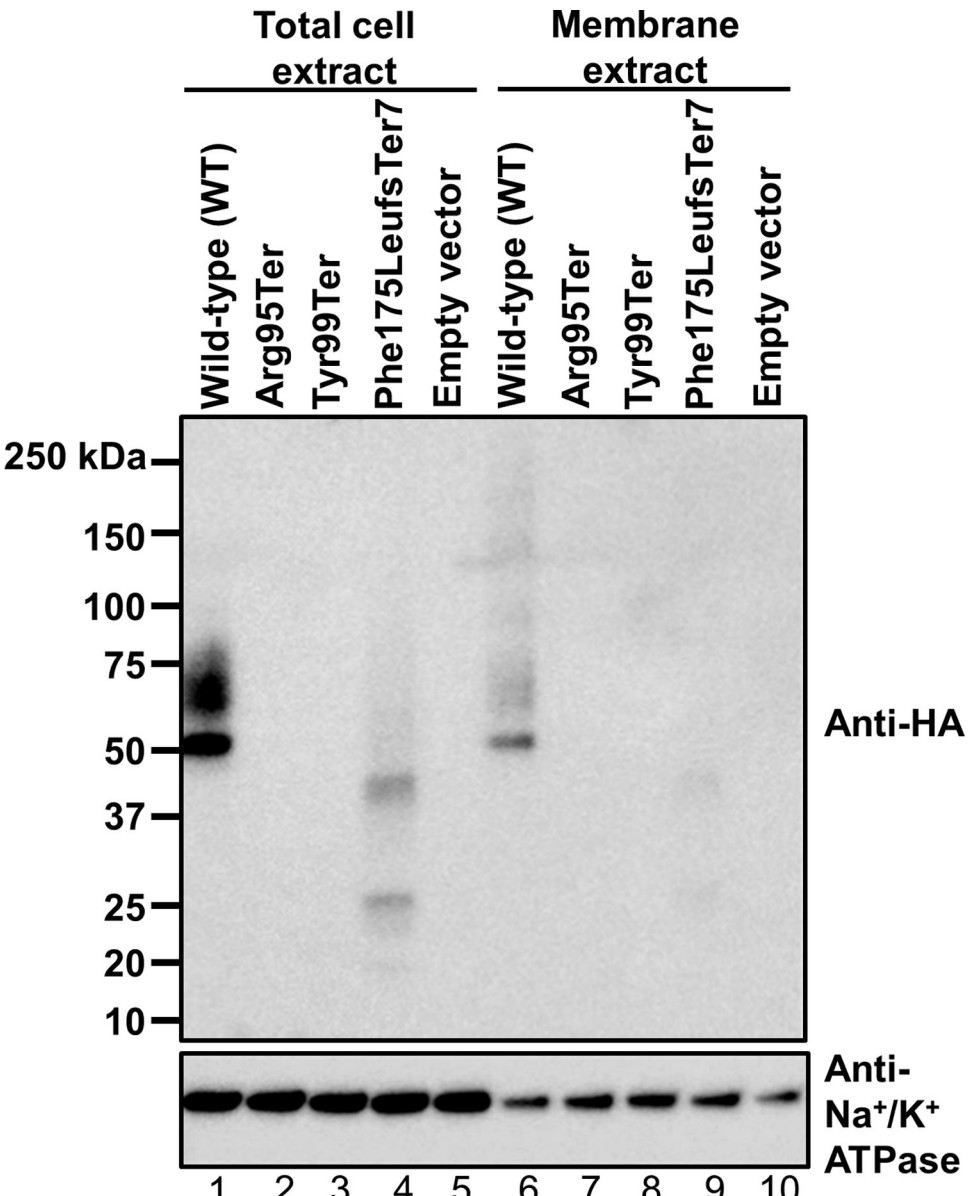

**Fig 1. GPR151 variant proteins are not stably expressed.** Western blot expression of HEK293 cells transfected with pcDNA3.1 plasmids encoding GPR151 variants with N-terminal HA-tag. The expected molecular weight of wild-type (WT) GPR151 is 47 kilodaltons (kDa). $Na^+/K^+$ ATPase is shown as a loading control.

2.2%), our study was adequately powered (80% at an $\alpha = 0.05$) to detect a mean difference of 0.36 kg/m$^2$ of BMI in knockouts compared to non-carriers. Similarly, for Tyr99Ter (aaf = 1.8%), our study was adequately powered (80% at an $\alpha = 0.05$) to detect a mean difference of 0.40 units of BMI in knockouts compared to non-carriers.

Unlike previous studies of *GPR151*, our burden tests and individual variant analyses failed to identify statistically significant associations with BMI in PGR (Table 1). Most importantly, homozygous *GPR151* knockout did not confer low BMI compared to non-carriers. We analyzed knockouts across all variants versus reference carriers and did not observe significant association for either the gene burden result (knockout n = 38, p = 0.98) or Tyr99X variant

**Table 1. *GPR151* associations with BMI.**

| GRCh38 chr: pos | Reference allele | Alternate allele | HGVSp | Genotype counts (RR\|RA\|AA) | P-value | Beta [95% CI] kg/m² (additive) | P-value (knockouts only) | Beta [95% CI] kg/m² (knockouts only) |
|---|---|---|---|---|---|---|---|---|
| 5:146515831 | G | A | Arg95Ter | 27273\|55\|1 | 0.82 | -0.126 [-1.23–0.98] | | |
| 5:146515817 | G | T | Tyr99Ter | 26350\|945\|34 | 0.92 | 0.0131 [-0.24–0.27] | 0.55 | 0.431 [-0.99–1.85] |
| 5:146515587 | CTA | C | Phe175LeufsTer7 | 27206\|120\|3 | 0.28 | 0.406 [-0.32–1.14] | | |
| Gene Burden | | | | 26150\|1141\|38 | 0.73 | 0.0405 [-0.20–0.28] | 0.98 | -0.021 [-1.37–1.33] |

chr, chromosome; pos, position; HGVSp, Human Genome Variation Society protein level change; R, reference allele; A, alternate allele; kg, kilograms; m, meter; CI, confidence interval

(n = 34, p = 0.55). We also performed a sample size-based meta-analysis with UK Biobank *GPR151* knockouts (n = 28) reported previously [28–30]. The meta-analyzed p-value remained non-significant (p = 0.67). In PGR, we also tested for associations with other relevant traits including waist-to-hip ratio, cholesterol and triglyceride levels and observed no significant associations or consistent trends (S2 Table).

As stated above, in PGR alone or in the combined meta-analyses, we did not observe a clinically meaningful effect on BMI in human knockouts despite a sizeable number of plof homozygous carriers for *GPR151*. We further examined if the previously reported weak genetic effect on BMI, largely conferred by heterozygous plof carriers, is reproducible. We meta-analyzed our results with summary statistics from the GIANT consortium [32] (Total N = 497,110; African Ancestry = 27,610; Admixed American Ancestry = 10,772; East Asian Ancestry = 8,839; European Ancestry = 449,889). With a significantly larger sample size, we replicated the Arg95Ter association with BMI (p = 6.72E-4; Beta = -0.042 [-0.063 –-0.0171]) with an additive model as used in the published study. There was no evidence of a population or study-specific effect (p-value for heterogeneity = 0.95). Hence, the original additive association between Arg95Ter and BMI, based primarily on heterozygous plof carriers, is reproducible but with a weak effect. Our findings with human knockouts across multiple plof variants indicate that complete absence of GPR151 does not further enhance this weak effect into a therapeutically meaningful reduction in BMI.

Next, we analyzed *GPR151* variants to assess reduction in T2D risk (Table 2). For gene-burden analyses, our study was powered to observe an odds ratio of T2D of 0.82 or lower (80% at

**Table 2. *GPR151* association with T2D.**

| GRCh38 chr: pos | Reference allele | Alternate allele | HGVSp | Genotypes cases (RR\|RA\|AA) | Genotypes controls (RR\|RA\|AA) | P-value | OR [95% CI] | P-value (knockouts only) | OR [95% CI] (knockouts only) |
|---|---|---|---|---|---|---|---|---|---|
| 5:146515831 | G | A | Arg95Ter | 6531\|30\|0 | 33106\|54\|2 | 0.24 | 1.85 [0.67–5.09] | | |
| 5:146515817 | G | T | Tyr99Ter | 6295\|265\|1 | 32008\|1115\|39 | 0.08 | 1.15 [0.98–1.35] | 0.99 | 0.99 [0.34–2.87] |
| 5:146515587 | CTA | C | Phe175LeufsTer7 | 6532\|29\|0 | 33003\|153\|6 | 0.87 | 1.03 [0.68–1.59] | | |
| Gene Burden | | | | 6234\|326\|1 | 31757\|1358\|47 | 0.03 | 1.18 [1.02–1.37] | 0.49 | 1.60 [0.42–6.0] |

chr, chromosome; pos, position; HGVSp, Human Genome Variation Society protein level change; R, reference allele; A, alternate allele; OR, odds ratio, CI, confidence interval

an α = 0.05). No individual plof variant was associated with a significant change in the risk for T2D. For the plof gene burden we observed a slight increase in T2D risk (P = 0.03, OR = 1.18 [1.02–1.37]), but this increase did not meet the threshold of significance corrected for multiple testing (threshold of p = 0.0083). Similar analyses for MI risk did not yield significant associations (S3 Table).

In total, complete loss of *GPR151* function in human knockouts was not associated with clinically meaningful changes in BMI or other traits related to obesity or metabolism.

## Body weight in *Gpr151⁻ᐟ⁻* mice

To determine if GPR151 plays a role in body weight regulation in mice, we generated *Gpr151* knockout (*Gpr151⁻ᐟ⁻*) mice using CRISPR-Cas9. In human, pig, and mouse, GPR151 is expressed primarily in the brain [27]. In mouse, *Gpr151* is expressed in the habenula, which is located in the dorsal thalamus of the brain [33]. In humans, *GPR151* is expressed in the midbrain [27]. In mice, we evaluated *Gpr151* expression by *in situ* hybridization (ISH). In wild-type (*Gpr151⁺ᐟ⁺*) mice, *Gpr151* mRNA expression was primarily observed in the habenular nucleus in the brain (Fig 2A), consistent with prior reports [33], and in mucosal cells of the ilium and jejunum (Fig 2C). *Gpr151* mRNA expression was not detected in *Gpr151⁻ᐟ⁻* mice (Fig 2B and 2D), confirming loss of expression in knockout animals.

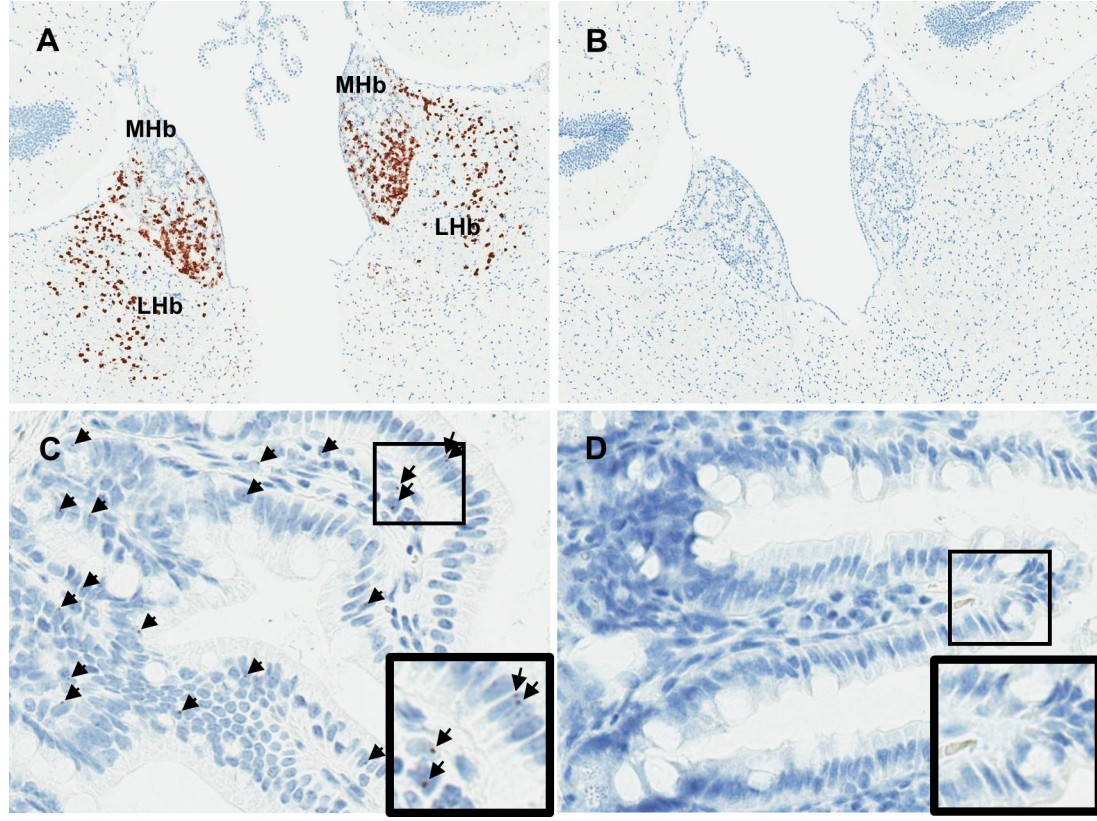

**Fig 2. *Gpr151⁻ᐟ⁻* mice do not express *Gpr151* mRNA.** (A and B) Sections of mouse brain containing the medial habenula (MHb) and lateral habenula (LHb) from *Gpr151⁺ᐟ⁺* and *Gpr151⁻ᐟ⁻* mice, respectively, stained with a riboprobe for *Gpr151*. (C and D) Sections of mouse intestine containing the ileum and jejunum from *Gpr151⁺ᐟ⁺* and *Gpr151⁻ᐟ⁻* mice, respectively, stained with a riboprobe for *Gpr151*. Black arrows indicate cells containing *Gpr151* mRNA. Inset shows higher magnification of boxed region.

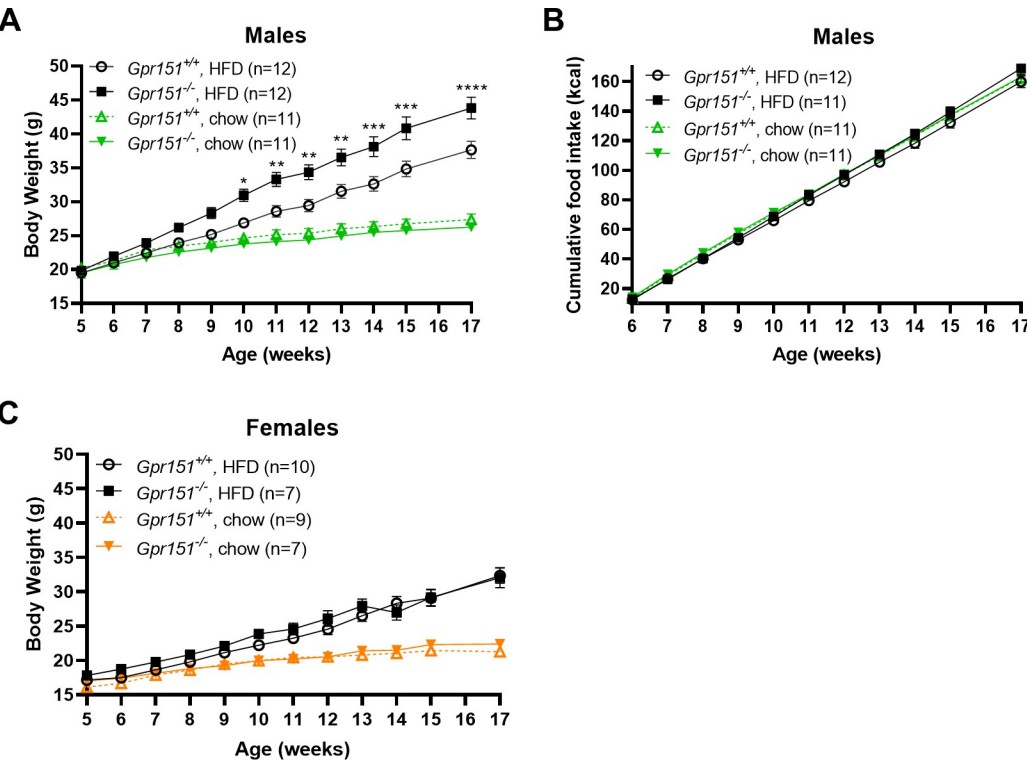

**Fig 3. Male *Gpr151*^-/- mice gain weight on high-fat diet (HFD).** (A) Body weights of male *Gpr151*^+/+ and *Gpr151*^-/- mice on a standard chow diet (chow) and high-fat diet (HFD). Data are presented as ± standard error of the mean (SEM). * p < 0.05, ** p < 0.01, *** p < 0.001, **** p < 0.0001, repeated measures, two-way ANOVA followed by post-hoc analysis using Sidak's multiple comparisons test. (B) Cumulative food intake in kcal (kilocalories) of male *Gpr151*^+/+ and *Gpr151*^-/- mice on a standard chow diet and high-fat diet. Data are presented as ± SEM. (C) Body weights of female *Gpr151*^+/+ and *Gpr151*^-/- mice on a standard chow diet and high-fat diet. Data are presented as ± SEM.

Next, we compared the body weights of *Gpr151*^+/+ and *Gpr151*^-/- on standard chow and a high-fat diet. On a standard chow diet, no difference in body weights was observed between *Gpr151*^-/- and *Gpr151*^+/+ control mice of either sex (Fig 3A and 3C). *Gpr151*^-/- male mice that were fed a high-fat diet for 12 weeks weighed ~16% more than *Gpr151*^+/+ controls (Fig 3A). Food intake was similar between the two groups suggesting an alternative mechanism for diet-induced weight gain in *Gpr151*^-/- male mice (Fig 3B). Female *Gpr151*^-/- mice that were fed a high-fat diet gained weight at the same rate as *Gpr151*^+/+ counterparts, suggesting a sex-based difference in the response to an obesogenic diet (Fig 3C).

In summary, complete loss of function of GPR151 is not associated with a clinically meaningful change (i.e., > 5% change) in BMI. We identified homozygous carriers of the previously published Arg95Ter plof and additional South Asia-specific plof variants in PGR, confirmed that plof variants are unstable *in vitro*, and observed no statistically significant reduction in BMI in either heterozygous or homozygous carriers. The body weights of male and female *Gpr151*^-/- mice were indistinguishable from *Gpr151*^+/+ control mice on a standard chow diet and were elevated in male *Gpr151*^-/- mice on a high-fat diet without a corresponding increase in food intake. The preclinical model data indicate that the lack of association with BMI is generalizable rather than a human population-specific phenomenon. Our results highlight the importance of taking into account the effect estimates and directionality of multiple loss-of-function variants when prioritizing GWAS results for functional follow-up. In aggregate, loss of GPR151 does not affect BMI in human knockouts in a clinically meaningful way and

GPR151 antagonism is likely not a compelling therapeutic strategy for BMI reduction or T2D remission in humans.

## Materials and methods

### Ethics statement

The Institutional Review Board (IRB) at the Center for Non-Communicable Diseases (IRB: 00007048, IORG0005843, FWAS00014490) approved the study. All participants gave written informed consent.

### Variant quality control (QC) and annotation

This study included a subset of 30,833 individuals with Whole Exome Sequencing and 9,292 individuals with Whole Genome Sequencing from the Pakistan Genome Resource (PGR). Samples were sequenced at an average of 30X coverage. Samples with low allele balance for a variant ($< 0.2$) or low depth ($< 10$) were set to missing and variants that had a missingness rate $> 5\%$ were removed. We also removed variants failing VQSR filters or failing visual validation on IGV. Variants were annotated using Variant Effect Predictor [34] based on the Ensembl101 gene model. For human *GPR151*, we used Ensembl Transcript ENST00000311104, which is the only annotated transcript for this gene. Variants annotated as frameshift, stop gained, splice acceptor and splice donor variants are considered plof variants. The three reported homozygous *GPR151* variants (Arg95Ter, Tyr99Ter, and Phe175LeufsTer7) have a call rate of 1 (i.e. zero missingness). Additionally, we filtered out plofs annotated as 'low confidence' according to the filtering criteria in LOFTEE [31]. Cumulative allele frequency (caf) was calculated as described [35].

### Case classification

Patients were categorized as T2D cases if they satisfied any one of the following criteria: (1) Physician diagnosis at a diabetes clinic, (2) HbA1c $> 6.5\%$, (3) use of glucose lowering medication or (4) fasting glucose $> 126$ mg/dl. An age of first diagnosis $>22$ years was used to exclude type 1 diabetes as much as possible. Patients were categorized as having had an MI as described previously [36].

### Statistical analysis

Associations with BMI (kg/m$^2$), cholesterol (mg/dl), triglycerides (mg/dl) and waist-to-hip ratio were analyzed using multivariate linear regression adjusting for age, sex, age$^2$ and top 5 genetic principal components (PCs). Associations with T2D and MI were analyzed using logistic regression, with Firth correction as implemented in glow [37]. The genomes and exomes datasets were analyzed separately and the summary statistics were meta-analyzed using inverse variance weighted meta-analysis as implemented in METAL [38]. Power calculations were performed using Quanto v1.2 [39]. To meta-analyze our results with the GIANT Consortium, we first transformed BMI values using rank-based inverse normalization. METAL was then used to perform inverse variance weighted meta-analysis.

### *In vitro* expression and western blot

Human codon optimized cDNAs, corresponding to GPR151 reference (wild-type; NCBI reference sequence NP_919227.2) or to nonsense mutant constructs, were cloned in pcDNA3.1(+) mammalian expression vectors with hemagglutinin (HA) epitope tags (YPYDVPDYA) appended to the amino termini. Transient transfection of adherent HEK293 cells was

performed using Lipofectamine 2000 (Invitrogen) in a 6-well plate format according to manufacturer's instructions. Cells were harvested 48–72 hours post-transfection by scraping, washed with phosphate-buffered saline (PBS), and pelleted by centrifugation at 300xg for 5 min. Whole-cell lysates were prepared from half of each sample by sodium dodecyl sulfate (SDS) extraction. Cells were resuspended in PBS containing 2.5% (weight/volume) SDS, samples were incubated at 4˚C for 10 minutes with end-over-end rotation, and insoluble material was removed by centrifugation at 16,000xg for 15 minutes. Membrane fractions were isolated from remaining cell sample using the Mem-PER Plus Membrane Protein Extraction Kit (Thermo-Scientific) according to manufacturer's instructions. Whole-cell lysates and isolated membrane fractions were analyzed by SDS polyacrylamide gel electrophoresis (PAGE) and Western Blot against the HA epitope to detect GPR151 expression. The following antibodies were used: anti-HA monoclonal antibody (Clone 2–2.2.14, Invitrogen 26183); anti-Na+/K+ ATPase alpha-1 antibody (clone C464.6, Sigma 05–369); and anti-mouse IgG HRP-conjugated antibody (R&D Systems HAF007).

## Generation and phenotyping of *Gpr151*$^{-/-}$ mice

Mice lacking the *Gpr151* gene were generated using the CRISPR-Cas9 system. All animal protocols were reviewed and approved by the Novartis Institutional Animal Care and Use Committee. The entire coding sequence of *Gpr151* is contained on a single exon (Ensembl gene ID# ENSMUSG00000042816). Two single guide RNA (sgRNA) sequences targeting sites just upstream of the translation start codon in exon 1 (ATCAAGCTCCTCCCTGCAGA) and within the 3' untranslated region (3' UTR) (TCATCAATATTGCTAAGCAG) were synthesized as crRNAs for Alt-R CRISPR-Cas9 system (Integrated DNA Technologies, Coralville, IA). A ribonucleoprotein mixture of the two crRNAs complexed with tracrRNA (Integrated DNA Technologies) and Cas9 protein (PNA Bio Inc, Newbury Park, CA) was electroporated into fertilized C57BL/6J embryos. The embryos were then implanted into pseudopregnant recipients. DNA lysates were prepared from tail biopsies of F0 generation pups using KAPA Mouse Genotyping Kit according to the manufacturer's instructions (Kapa BioSystems, Cat# KK7302). Mice were genotyped by polymerase chain reaction (PCR) using the following primers: For1 (5'-ACTTACAGACACTGTGAACAGC-3') anneals to sequence upstream of *Gpr151* exon 1, For2 (5'-TGGCTCCCAGAGTGGATAGC-3') anneals to sequence within exon 1, and Rev1 (5'-TGCCTTTCTACTTACCAGGTTC-3') anneals to sequence downstream of the Cas9 cut site within the 3' UTR. For2 and Rev1 amplify a product of 614 bp corresponding to the wild-type allele, and For1 and Rev1 amplify a product of ~233 bp corresponding to the null allele. PCR conditions were as follows: denaturation at 95˚C for 3 min, 35 cycles of 15 sec at 95˚C, 15 sec at 60˚C, and 30 sec at 72˚C, then 5 min at 72˚C. F0 founders were bred to C57BL/6J mice for germline transmission of mutant alleles. The null allele of the F1 founder line selected was confirmed by Sanger sequencing (GeneWiz, South Plainfield, NJ) to have a deletion of 1343 bp between the expected Cas9 cleavage sites. Heterozygous mice were interbred to generate homozygous offspring for studies.

At five weeks of age, male *Gpr151*$^{+/+}$ and *Gpr151*$^{-/-}$ littermates were split from group housing to individual housing to monitor body weight and food consumption. Female mice remained group housed for body weight studies. At five weeks of age, all animals were provided either a standard chow diet (Purina Picolab 5053) or a high-fat diet deriving 60% kcal from fat (Research Diets D12492i), with *ad libitum* access to food and water. Body weights (male and female) and food intake (males only) were measured 1–2 times per week for 12 weeks following exposure to high-fat diet. Animals were maintained on a 12-hour light/dark cycle. The study size is shown in Fig 3. One cohort was run for the study. In Fig 3B, one animal

in the *Gpr151*^-/- HFD group was excluded from analysis because food intake data past week 8 was lost. Raw data are shown in S4 Table.

After 12 weeks of study, the mice were euthanized and brains collected to confirm *Gpr151* genotype by *in situ* hybridization. Samples from three animals of each genotype were used. Whole brains were fixed for 48 hours in 10% neutral buffered formalin and later embedded in paraffin. Four microns sections of each brain were collected. Staining was performed on the Leica Bond RX automated staining platform using the RNAscope 2.5 LSx Reagent Kit (ACD-Bio, Bio-Techne– 322440) with a mouse *Gpr151* probe (ACDBio, Bio-Techne—317328) following the standard RNAscope Assay [40].

## Supporting information

**S1 Table. List of *GPR151* plof variants identified in 30,833 exomes and 9,292 genomes** (DOCX)

**S2 Table. *GPR151* association with waist-to-hip ratio and lipid-related biomarkers** (DOCX)

**S3 Table. *GPR151* association with MI** (DOCX)

**S4 Table. Underlying numerical data for Fig 3** (XLSX)

## Author Contributions

**Conceptualization:** Allan Gurtan, John Dominy, Daniel Denning, Diana Shpektor, Danish Saleheen.

**Data curation:** Shareef Khalid, Treeve Currie, Asif Rasheed, Shahid Hameed, Subhan Saeed, Imran Saleem, Anjum Jalal, Shahid Abbas, Raffat Sultana, Syed Zahed Rasheed, Fazal-ur-Rehman Memon, Nabi Shah, Mohammad Ishaq, Philippe Frossard, Danish Saleheen.

**Formal analysis:** Shareef Khalid, Anastasia Gurinovich.

**Funding acquisition:** Danish Saleheen.

**Investigation:** Allan Gurtan, John Dominy, Shareef Khalid, Linh Vong, Shari Caplan, Treeve Currie, Sean Richards, Lindsey Lamarche, Anastasia Gurinovich, Danish Saleheen.

**Methodology:** Allan Gurtan, John Dominy, Shareef Khalid, Linh Vong, Shari Caplan, Treeve Currie, Sean Richards, Lindsey Lamarche, Anastasia Gurinovich, Philippe Frossard, Danish Saleheen.

**Project administration:** Allan Gurtan, John Dominy, Asif Rasheed, Imran Saleem, Nabi Shah, Danish Saleheen.

**Resources:** Shari Caplan, Amit V. Khera, John Danesh, Danish Saleheen.

**Software:** Shareef Khalid, Anastasia Gurinovich.

**Supervision:** Allan Gurtan, John Dominy, Linh Vong, Asif Rasheed, Shahid Hameed, Anjum Jalal, Shahid Abbas, Raffat Sultana, Syed Zahed Rasheed, Fazal-ur-Rehman Memon, Mohammad Ishaq, Danish Saleheen.

**Validation:** Shareef Khalid.

**Visualization:** Linh Vong, Treeve Currie, Sean Richards, Lindsey Lamarche.

**Writing – original draft:** Allan Gurtan, John Dominy, Shareef Khalid, Danish Saleheen.

**Writing – review & editing:** Allan Gurtan, John Dominy, Shareef Khalid, Linh Vong, Shari Caplan, Treeve Currie, Sean Richards, Lindsey Lamarche, Daniel Denning, Diana Shpektor, Anastasia Gurinovich, Asif Rasheed, Shahid Hameed, Subhan Saeed, Imran Saleem, Anjum Jalal, Shahid Abbas, Raffat Sultana, Syed Zahed Rasheed, Fazal-ur-Rehman Memon, Nabi Shah, Mohammad Ishaq, Amit V. Khera, John Danesh, Philippe Frossard, Danish Saleheen.

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
