## [Decision Letter · Decision Letter 0]

27 Nov 2021

Dear Dr Gurtan,

Thank you very much for submitting your Research Article entitled 'Analyzing human knockouts to validate GPR151 as a therapeutic target for reduction of body mass index' to PLOS Genetics.

The manuscript was fully evaluated at the editorial level and by independent peer reviewers. The reviewers appreciated the attention to an important problem, but raised some substantial concerns about the current manuscript. Based on the reviews, we will not be able to accept this version of the manuscript, but we would be willing to review a much-revised version. We cannot, of course, promise publication at that time.

If you decide to revise the manuscript for further consideration at PLOS Genetics, please aim to resubmit within the next 60 days, unless it will take extra time to address the concerns of the reviewers, in which case we would appreciate an expected resubmission date by email to plosgenetics@plos.org.

[LINK]

We are sorry that we cannot be more positive about your manuscript at this stage. Please do not hesitate to contact us if you have any concerns or questions.

Yours sincerely,

Giles S. H. Yeo

Guest Editor

PLOS Genetics

Gregory Barsh

Editor-in-Chief

PLOS Genetics

It is clear that all three reviewers have found the work of interest. However, they (in particular the second reviewer) raise some important issues, that I agree should be addressed, that would improve this manuscript. This piece of work, if true, is an important 'negative', and should hopefully set a precedent for future similar studies. So I believe that the additional rigorous analyses suggested should be undertaken.

Reviewer's Responses to Questions

**Comments to the Authors:**

Reviewer #1: GPR151 is a gene implicated as a GWAS hit in obesity and considered as a possible drug target for lowering BMI. In previous reports, a nonsense variant was associated with lower BMI, suggesting inhibition could be therapeutically beneficial. The authors provide multiple lines of evidence that, unfortunately, show that this gene is not a good drug target for BMI. The truncating variants considered here really do abolish protein expression in cell culture, and yet, association of truncating variants with lower BMI does not replicate in a Pakistani population, no obvious difference in BMI is observed even in homozygous knockouts in this cohort enriched for autozygosity, and no obvious difference in BMI is observed in knockout mice. Indeed, a couple of observations — higher BMI in male mice fed a high fat diet, and nominally (P=0.03) higher T2D risk in people with truncating variants — point in the opposite direction.

This is a careful and thorough paper on an important topic. Functional analyses of GWAS hits like these are vitally important and the results reported here will probably save millions of dollars in wasted preclinical drug discovery efforts. The science is well done and paper well-written, I have no major issues with anything here. A few minor points follow:

In the introductory paragraph beginning “G protein coupled receptors (GPCRs) are attractive drug targets…” it might be nice to give some background for the reader on anything known about GPR151 in particular. I looked it up in GTEx and was surprised to see it expressed at appreciable levels only in brain. Perhaps none of this matters since you found it’s not associated with BMI anyway, but for me as an outsider who has never thought about this gene or this phenotype before, any background you could give would be helpful.

I kept looking for evidence that the protein is not expressed in the KO mice. Then I realized that the use of HA tag in the in vitro experiments and the use of a riboprobe in the mouse sections must be because there simply exists no good antibody for this protein. If so, this fact might be stated up front to avoid the reader wondering. A quick google search indicates some vendors claim to sell GPR151 antibodies; if the authors already tested these and found them not to work, that info might be useful to some readers.

I found myself wondering how the originally reported association could so spectacularly fail to replicate. Two possibilities came to mind. One, the UKBB association was right at the edge of significance (4.9e-8 in Emdin 2018, 5.7e-9 in Akbari 2021) — was it just a false positive? Two, is there any chance that there is some real effect on BMI but only through an interaction with age or some other variable? How does the median age in this cohort compare to that in UKBB? Might the KO mice have ended up with lower BMI at some age greater than 17 weeks? I agree with the authors that the data already presented here are pretty much sufficient to kill any interest in this gene as drug target, but some discussion/caveats on this topic might be nice to have.

Eric Vallabh Minikel

November 8, 2021

Reviewer #2: In this manuscript the authors were investigating whether they could replicate previous associations between rare putative loss of function (pLOF) variants in GPR151 and obesity/BMI, and type 2 diabetes risk. To do this they analysed sequence data from the Pakistan Genome Resource PGR, including 30,833 individuals with whole-exome data and 9,292 with whole genome sequence data. The authors identified three pLOF, the previously described Arg95Ter, and two additional variants Tyr99Ter and Phe175LeufsTer7, they tested the effect of these variants individually and in a gene burden test for association with BMI and a number of additional related traits (5 additional in total). The authors also investigated in vitro the effect of these variants on expression and created a knockout mouse which they studied under chow and high fat diet conditions. Overall the authors conclude from their analyses that there is no compelling evidence that targeting GPR151 with antagonists would be an effective approach for obesity treatment.

Overall the manuscript is clear, concise and makes an interesting point. However there are some areas of concern:

1. Abstract – The authors state “Moreover, loss of GPR151 confers a nominally significant increase in risk of T2D (odds ratio = 1.2, p value = 0.03). Relative to wild-type mice, Gpr151-/- animals exhibit no difference in body weight on normal chow, and higher body weight on a high-fat diet, consistent with the findings in humans.” Firstly, the authors state the association with increased risk of T2D is nominally significant but this does not take into account the number of tests done (BMI, T2D, cholesterol, triglycerides, waist-hip ratio, MI) so I think this result as stated risks over-interpreting the data (additional comments on T2D and other analysis in other points below). Secondly, the authors state that knockout mice have higher body weight on a high-fat diet, and that this is consistent with human findings. But this is confusing as the initial human data suggested that lof variants in GPR151 lowered BMI in humans and the authors here with their own data show that they do not see any evidence for differences in BMI in humans with lof, so unclear what is meant by being consistent with findings in humans.

2. Background/ context of the findings - Some of the introduction is missing out key recent papers such as Sobreira et al., 2021 relating to the effects at the FTO locus in nearby loci. Also the statement regarding melanocortin 4 receptor agonists is somewhat misleading. Though many of these compounds do have undesirable cardiovascular effects this is not true of all, and so the phrase should be re-stated. Setmelanotide seems to be well tolerated with minimal side effects, this is published in e.g. Clement et al 2018; Haw et al., 2020 and indeed the review cited by the authors Yeo et al 2021 shows this clearly in table 3. I think it is still valid to state additional body weight reduction drugs are needed but important to ensure the statements made are correct.

3. An important point which the authors do not make is that even in the original publications describing an association between pLOF variant Arg95Ter, the effect on BMI was very modest (−0.36 kg/m2). So arguably, the expectation would be that this effect would be too modest to have meaningful clinical impact, although I acknowledge the previous papers were mostly focused on additive effects. However, there are 20 homozygous carriers for this variant in biobank (and only one is the authors data) and the authors could easily investigate this and combine it in meta-analysis with their own data, which I do think is warranted to gain clarity as to the effect of this variant on BMI.

4. Critically, I think there are two questions that the authors need to consider separately:

a. Is there evidence for replication/lack of replication of an association between the previous pLOF variant (Arg95Ter) at this locus and reduced BMI? I note that here the authors show that their results are still consistent (CIs overlap) with a protective effect of this variant on BMI (their numbers here are smaller than previously published so lack power). I would suggest a meta-analysis of this variant across all available cohorts with this data is warranted as the data shown here do not provide evidence “against” this association. The authors might consider including data from non overlapping previous datasets and if possible the Genes and Health initiative as another effort enriched for autozygous individuals.

b. Is there evidence that complete loss of function at this locus will have a clinically meaningful effect on BMI in humans?

The two questions are not exactly identical because although the authors do not have evidence for a clinically meaningful reduction in BMI in their population, their data do not refute an association between the Arg95Ter and reduced BMI. Indeed their CIs overlap previous effect estimates with larger sample sizes. So I think a meta-analysis across all datasets with this variant is warranted to try and establish whether the original association stands, or not.

Regarding the second point one might argue even if the effects replicate the effect on BMI overall is modest. Again I think given the available data in UK biobank including additional homozygous pLOF carriers this point would be best addressed by meta-analysing the results across all possible datasets the authors can access, they clearly have access to UK biobank so this should be straightforward. I’d suggest if possible including data from Genes and Health would also be interesting and add value. Importantly if the desire is to include only data from null alleles it would be critical to ensure incomplete loss of function variants are not included in the burden test.

5. Looking at the data in Table 1, the second termination variant also has an effect size point estimate that is consistent with lower BMI for homozygous carriers. So I think there is a real question whether the frameshift variant which occurs much later in the protein is fundamentally different. Indeed the author’s data show that this variant is expressed although at low levels. The authors conclude this variant is loss of function because of its lower levels of expression but two bands are clearly seen so the variant is expressed, and there is no in vitro functional data to support the statement that this is a complete loss of function variant. Given this, it would be good to see the gene burden test results removing this variant from the burden test.

6. The association with T2D is nominal only and not adjusted for the different tests and phenotypes looked at, so I think interpretation needs to take this into account. Specifically, and if the authors remove the frameshift variant it looks like their CIs overlap the previous estimates for a protective effect? Instead of a straight power calculation for what the authors are powered to detect, I would prefer to see a power calculation for what effect sizes the authors are powered to rule out? Again I think combining this new data with previously published data in meta-analysis would increase power and provide more clarity as to what the data are showing.

7. The mouse data suggest a possible sexual dimorphism in the phenotype of the knockout mice, have the authors analysed the human data stratified by sex? I think despite smaller numbers and loss of power it would be interesting to check whether there is any evidence from human data for different variant effects between the sexes.

8. The manuscript is missing a discussion on how the authors interpret their results in light of previous association results with reduced BMI and obesity at this gene in much larger sample sizes (including comparable numbers for some homozygous individuals and variants)? Specifically, in three different cohorts a burden of pLOF and missense predicted deleterious variants associated with reduced BMI. How do the authors interpret their data in light of previous findings? I think a meta-analysis across available datasets may help provide further clarity here.

9. Data availability: I could not find a specific data availability statement in the manuscript aside from a “no-some restrictions will apply” in the box at the front. Please clarify exactly what data will be available and how, for example is the mouse line available from somewhere, will the summary statistics for all GPR151 variants and associated phenotypes analysed in the manuscript be available somewhere? Although the authors mention all associated data is within the manuscript this is not really the case as full genotype counts for ref/ref ref/alt and alt/alt and corresponding phenotypes are not given for every phenotype tested. If some data have restricted access please explain what data cannot be made available and why.

Minor issues:

1. In all the tables, for clarity it would be helpful to see the N total in cases / controls or in the entire test data not just hets and hom carriers, or better still the number of each genotype class in cases and controls separately.

2. There is no author summary provided.

3. Figure 2 - I suggest modifying Figure 2 title to better represent entire multipanel figure or pulling out the weight curves into separate figure. Would encourage authors to change the colour from male and female mice away from stereotypes of blue and pink.

4. Figure 2D the riboprobe for GPR151 intestine data in wild-type is not particularly obvious and its detection seems a little subjective.

5. Methods: “We obtained a list of high-quality protein coding transcripts with annotated start and stop codons.” Please clarify where this was obtained from or how exactly is a high-quality protein coding transcript defined?

6. Methods: Case classification, T2D cases “1) Documented history of diabetes” , please specify what documented history of diabetes means? Also, how was type 1 diabetes, excluded?

7. Methods: adjusted for top 5 principle components, why 5?

8. Unclear whether genomes and exomes were treated the same or whether there was any adjustment for batch effect in the analysis? It looks like the data were analysed separately and then meta-analysed but it would be good to make this a bit clearer.

9. Methods : somewhat unclear the authors state that “At six-weeks of age, Gpr151-/- mice and wild type littermates were individually housed for body weight and food consumption measurements and provided either a standard chow diet (Purina Picolab 5053) or high fat diet in which 60% kcal is derived from fat (Research Diets D12492i) with ad libitum access to water.” But in the following sentence it is stated that female mice were group housed, so were they group housed from the outset and then from six-weeks the male Kos and wild-type only were individually housed?

Reviewer #3: The present article by Gurtan et al. used the Pakistan Genome Resource, which includes high rate of human homozygous loss-of-function (KO) due to high consanguinity, to validate GPR151 as a potential drug target. Despite accurate statistical power, the authors did not find any significant association between three loss-of-function GPR151 variants and BMI. The authors also investigated mouse models deleted for Gpr151 and found that these mice had no difference in body weight in normal chow.

This article is well written and the reviewer believes that these negative results are of great interest for the community (it is crucial to publish negative results).

The reviewer has the following comments:

- If the reviewer is right, the authors actually combined carriers of heterozygous and homozygous LOF GPR151 variants for their burden analysis, while the abstract and introduction mainly tackled “human homozygous loss-of-function”. It would be important to also analyze carriers of homozygous LOF GPR151 variants only (and remove the carriers of heterozygous variants). Furthermore, to enhance the statistical power of this analysis, the authors could combine the carriers of the three LOF variants (at homozygous state).

- The mean depth of coverage of GPR151 in the Pakistan Genome Resource should be provided, as well as the genotyping success rate for the three variants.

- The authors should also analyse the effect of LOF variants (in homozygous carriers) on obesity risk (obese participants versus normal weight participants)

**Have all data underlying the figures and results presented in the manuscript been provided?**

Reviewer #1: **No: **There is currently no "data availability" statement in the manuscript

Reviewer #2: Yes

Reviewer #3: Yes

PLOS authors have the option to publish the peer review history of their article (what does this mean?). If published, this will include your full peer review and any attached files.

Reviewer #1: **Yes: **Eric Vallabh Minikel

Reviewer #2: No

Reviewer #3: **Yes: **Amelie Bonnefond

---

## [Decision Letter · Decision Letter 1]

13 Feb 2022

Dear Dr Gurtan,

We are pleased to inform you that your manuscript entitled "Analyzing human knockouts to validate GPR151 as a therapeutic target for reduction of body mass index" has been editorially accepted for publication in PLOS Genetics. Congratulations!

Yours sincerely,

Giles S. H. Yeo

Guest Editor

PLOS Genetics

Gregory Barsh

Editor-in-Chief

PLOS Genetics

Comments from the reviewers (if applicable):

The reviewers are all now enthusiastic about accepting this manuscript for publication, and I agree. Please just note the remaining minor comments from reviewer #2.

Reviewer's Responses to Questions

**Comments to the Authors:**

Reviewer #1: My concerns have been adequately addressed.

Reviewer #2: The authors have addressed my concerns and I think the manuscript is considerably improved. However there are a few minor issues that still merit stating clearly for accuracy.

1. The authors have predicted a full loss of function for Phe175LfsTer7 based on computational prediction but the experimental in vitro evidence does not provide proof that this variant is a complete loss of function. Since no functional readouts aside from protein expression were done, I think the wording should be clear on this point. For clarity the authors should amend the sentence “The severity and diminished expression of this truncation indicate that this variant is a loss-of-function.” To say “suggest” instead of “indicate”. I agree that given the nature of the predicted truncation this is very likely to be loss of function but in theory it could be dominant negative so I think “suggest” would be more appropriate wording.

2. A minor point is that a diabetes age of diagnosis after age 22 is not a guarantee that this does include a small proportion of patients with type 1 diabetes. See for example, doi: 10.1016/S2213-8587(17)30362-5.

Reviewer #3: The authors accurately answered to my comments

**Have all data underlying the figures and results presented in the manuscript been provided?**

Reviewer #1: Yes

Reviewer #2: Yes

Reviewer #3: None

PLOS authors have the option to publish the peer review history of their article (what does this mean?). If published, this will include your full peer review and any attached files.

Reviewer #1: **Yes: **Eric Vallabh Minikel

Reviewer #2: No

Reviewer #3: **Yes: **Amélie Bonnefond

**Data Deposition**

http://datadryad.org/submit?journalID=pgenetics&manu=PGENETICS-D-21-01413R1

**Press Queries**

---

## [Editor Report · Acceptance letter]

31 Mar 2022

PGENETICS-D-21-01413R1 

Analyzing human knockouts to validate GPR151 as a therapeutic target for reduction of body mass index 

Dear Dr Gurtan, 

We are pleased to inform you that your manuscript entitled "Analyzing human knockouts to validate GPR151 as a therapeutic target for reduction of body mass index" has been formally accepted for publication in PLOS Genetics! Your manuscript is now with our production department and you will be notified of the publication date in due course.

With kind regards,

Katalin Szabo

PLOS Genetics

On behalf of:
